# Sparse Convolutional Denoising Autoencoders for Genotype Imputation

**DOI:** 10.3390/genes10090652

**Published:** 2019-08-28

**Authors:** Junjie Chen, Xinghua Shi

**Affiliations:** Department of Bioinformatics and Genomics, College of Computing and Informatics, University of North Carolina at Charlotte, 9201 University City Blvd, Charlotte, NC 28223, USA

**Keywords:** genotype imputation, convolutional neural network, autoencoder, sparse model, deep learning

## Abstract

Genotype imputation, where missing genotypes can be computationally imputed, is an essential tool in genomic analysis ranging from genome wide associations to phenotype prediction. Traditional genotype imputation methods are typically based on haplotype-clustering algorithms, hidden Markov models (HMMs), and statistical inference. Deep learning-based methods have been recently reported to suitably address the missing data problems in various fields. To explore the performance of deep learning for genotype imputation, in this study, we propose a deep model called a sparse convolutional denoising autoencoder (SCDA) to impute missing genotypes. We constructed the SCDA model using a convolutional layer that can extract various correlation or linkage patterns in the genotype data and applying a sparse weight matrix resulted from the *L*_1_ regularization to handle high dimensional data. We comprehensively evaluated the performance of the SCDA model in different scenarios for genotype imputation on the yeast and human genotype data, respectively. Our results showed that SCDA has strong robustness and significantly outperforms popular reference-free imputation methods. This study thus points to another novel application of deep learning models for missing data imputation in genomic studies.

## 1. Introduction

Genotype imputation is a critical step in many types of genomic analysis, ranging from genome wide association studies (GWAS) to phenotype prediction. Missing values in genotype data are common and could result from many reasons such as low call rates, deviations from Hardy–Weinberg equilibrium, and the abundance of rare or low frequent variants in samples [1,2]. Genotype imputation works by computationally inferring missing values in a genotype profile, typically using the correlation or linkage information from untyped variants and nearby markers that are genotyped [3,4]. Genetic variants whose genotypes are imputed are mostly single nucleotide polymorphisms (SNPs), although other types of genetic variants such as small insertions and deletions and large structural variants can be imputed as long as they are in linkage disequilibrium (LD) with typed variants [5,6].

Existing imputation methods can be generally classified into two categories based on whether a reference panel is required, as summarized in Table 1. Methods in the first category require a reference panel, which contains haplotype information from many samples usually from the same or similar population background. These reference-based imputation methods usually apply a haplotype-clustering algorithm [7] and a hidden Markov model (HMM) [8] to impute missing SNPs using known haplotypes as a reference [9], such as those from the HapMap Project [10] or the 1000 Genomes Project [5,6] for human genomes. Particularly, fastPHASE [7] uses a localized haplotype-clustering model, in which reference haplotypes are grouped into clusters at each SNP for imputing missing genotypes at that locus. IMPUTE [8] is based on an extension of the HMM algorithm originally developed as part of the importance sampling scheme for simulating coalescent trees, modelling LD, and estimating recombination rates. IMPUTE2 [11] is a flexible and scalable extension of the original IMPUTE algorithm [8]. It uses an adaptive haplotype selection approach to impute untyped SNPs in a linear time (*O*(*N*)) compared with the quadratic time (*O*(*N*^2^)) carried out in IMPUTE [8]. MACH [12] uses an HMM model that works by successively updating the phase of each individual’s genotypes conditional on the current haplotype estimates of all other individuals. Minimac4 [13] is the latest version in a series of genotype imputation software-preceded by Minimac3 (2015) [14], Minimac2 (2014) [15], minimac (2012) [16], and MaCH (2010) [12]. Minimac4 [13] is a lower memory demanding and a more computationally efficient implementation of the original MACH algorithms with comparable imputation quality. BEAGLE [17,18] is another common imputation tool based on a graphical model of a set of haplotypes. It works iteratively by fitting the model to the current set of estimated haplotypes and then resampling new estimated haplotypes for each individual using a fitted model. The probabilities of missing genotypes are calculated from the model that is fitted at the final iteration. Additionally, SNP tagging-based approaches such as PLINK [19], SNPMSTAT [20], and TUNA [21] carry out genotype imputation using LD information on tag SNPs [22,23]. Specifically, for each SNP to be imputed, the reference haplotypes are used to search for a small set of tag SNPs in the flanking region that forms a local haplotype background in high LD with the target SNP to be imputed. All of the methods aforementioned rely on a well-defined haplotype-reference panel and would not work for those species without a high-resolution reference panel.

Instead, a suite of methods for missing data imputation that do not require a reference panel are developed based on statistical inference of unobserved data, which can be utilized for genotype imputation. These methods use information such as the row average approach [24], distance or similarity based methods like the k-nearest neighbors (KNN) algorithm [25], singular value decomposition (SVD) [26], and prediction models based on classification or regression [26,29]. Specifically, as one of the simplest methods, the row average approach [24] imputes missing values using an average of all the non-missing values or the most frequent value in the same column with missing values. Distance or similarity based methods usually exploit data that are similar or close to the missing data to make inference of missing values. The commonly used similarity measurements include Pearson correlation, Euclidean distance, variance minimization, and cosine distance. For example, the KNN imputation algorithm [25] finds the *k*-nearest neighbors that have values in the missing value positions, and uses a weighted average of the values from *k*-nearest neighbors to estimate missing values. Evidence has shown that log-transformation can sufficiently reduce the effect of outliers on similarity measurements [30] and can thus be performed before imputing missing data using distance or similarity based methods. An SVD-based imputation method [26] obtains the *k* most significant eigenvectors to impute the missing values using a low-rank SVD approximation estimated by an expectation maximization (EM) algorithm. Prediction-model-based imputation methods [26,29] create a predictive model to estimate values that will substitute missing data in a machine learning fashion. In this case, the input data are divided into two sets: one set with no missing values for training and the other set with missing values for testing. The predictive models can be trained using popular regression or classification models such as logistic regression [27,31] and random forest [28,32].

Recently, deep learning [33] has shown great potential in numerous applications including image processing [34,35], voice recognition [36,37], natural language processing [38,39], and particularly bioinformatics [40]. Applications of deep learning in Bioinformatics include variant calling [41], functional annotation [42,43], protein structure recognition and prediction [44,45,46,47,48,49,50], gene expression inference [51], molecular function recognition [52], prediction of methylation states [53], and high-throughput chromosome conformation capture (HiC) data enhancement [54]. Deep learning-based methods, especially autoencoders, have been reported to work well to address the missing data problems in various fields [55,56]. For instance, autoencoders have been applied to impute missing data in electronic health records [55] and human immunodeficiency virus (HIV) data [57]. Another example is a multiple-layer perceptron-based denoising autoencoder method for imputing DNA methylation data with comparable performance with the SVD approach [58]. However, the commonly used autoencoder architectures are based on fully connected layers in which each neuron is connected to every neuron in a previous layer, and each connection has its own weight. Learning on this fully connected architecture is very expensive in terms of computational time and space. Furthermore, fully connected autoencoders ignore the underlying structure or relationship in genomic data such as the LD structure in genotype profiles. Therefore, the limitations of the current practice of deep learning methodology in genomic analysis leave a vast room for model improvement, especially for those models based on the autoencoder framework. One particular technique to encode data relatedness or correlation is to use convolutional networks. A convolutional network can learn the underlying structure and relationship in genotype data by leveraging a convolutional kernel that is capable of learning various local patterns in a filter window. To handle high dimensional genomics data where the feature size is significantly larger than the sample size, we can introduce model sparsity by incorporating regularization on the weight matrix of a deep learning model.

Hence, in this study, we propose a novel deep learning model, called sparse convolutional denoising autoencoder (SCDA), for genotype imputation that does not need to compare with a reference panel. Specifically, the SCDA model utilizes convolutional layers to take account of local data correlations in the general autoencoder framework, and incorporates model sparsity to handle high dimensional genomic data using an *L*_1_ regularization on each convolutional kernel. To comprehensively evaluate the performance of the SCDA model for genotype imputation, we simulated different missing data scenarios on a yeast genotype dataset and a human leukocyte antigen (HLA) genotype dataset, respectively. Our results showed that SCDA achieved higher imputation accuracy than three existing reference panel-free imputation methods, and demonstrated the strong robustness of this SCDA model on different missing data scenarios. SCDA’s nice performance for genotype imputation benefits from using convolutional layers to extract linkage patterns in the genotype data and a sparse weight matrix resulted from the *L*_1_ regularization to handle high dimensional data. This study thus demonstrates a new application of deep learning models to impute missing data in genomic studies.

## 2. Materials and Methods

### 2.1. Dataset

To evaluate our proposed SCDA method, we used a comprehensively assayed yeast genotype dataset [59] and a human genotype dataset from the most extensive catalog of human genetic variation from the 1000 Genomes Project [5,6]. The yeast data represents a scenario that the genetic background is simple and the genotypes are highly correlated. The human data represent a more realistic and complex scenario where the genotypes are sampled from diverse human populations. The yeast genotype dataset contains the genotype profile of 28,820 unique genetic variants, which was obtained by sequencing 4390 observations from a cross between two strains of yeast: a widely used laboratory strain (BY) and an isolate from a vineyard (RM). The original data fields in the yeast genotype profile were encoded as -1 for BY and 1 for RM. As the loss function in our SCDA model requires non-negative data fields, we replaced all -1 values with 2 in data preprocessing. For the human genotype data, we chose to impute genotypes of the human leukocyte antigen (HLA) [60]. As the HLA region contains a gene complex encoding the major histocompatibility complex (MHC) proteins in humans, HLA represents a region where genotypes are diverse, heterogeneous, and complicated. We extracted the HLA genotype data consisting of 27,209 unique genetic variants in 2504 individuals across five super populations worldwide sequenced from the 1000 Genome Project [5], including Americans (AMR), Southern Asians (SAS), East Asians (EAS), Europeans (EUR), and Africans (AFR). The EAS super population consists of 617 individuals from six populations, including Chinese Dai in Xishuangbanna (CDX), Han Chinese in Beijing (CHB), Chinese in Denver (CHD), Southern Han Chinese (CHS), Japanese in Tokyo (JPT), and Kinh in Ho Chi Minh City (KHV). The human genotypes are encoded as 1 for the original genotype of ‘0|0’, 2 for ‘0|1’ or ‘1|0’, and 3 for ‘1|1’, respectively, converting from the original variant call format (VCF) file of the 1000 Genomes Project. We can see that both the yeast and human genotype datasets are highly dimensional with the feature dimension (*p* = 28,820 for yeast genotypes, *p* = 27,209 for HLA genotypes) significantly larger than the sample size (*n* = 4390 for yeast, *n* =2504 for HLA). With this type of highly dimensional dataset, sparse models that use regularization to impose sparsity work well to address the problem of the curse of dimensionality [61].

In order to assess the performance of our SCDA method in different missing data scenarios, we generated three sets of synthetic datasets by randomly masking 5%, 10%, and 20% of the original genotypes to zeros in the original yeast and human HLA datasets, respectively. For each of these synthetic datasets, we split the data into three separate datasets containing 65%, 15%, and 20% of the synthetic data for training, validation, and testing, respectively. Figure 1 visualizes data vectors of two samples with 5% missing values randomly selected from the yeast genotype dataset (Figure 1a) and HLA genotype data (Figure 1b). Different colors represent different genotypes and missing values are denoted in white color. Consecutive color blocks indicate that genotypes are highly correlated among nearby genetic markers, resulting from LD and linkage patterns. As expected, the correlations among yeast genotypes are much stronger than those among HLA genotypes. Hence, compared with the yeast data, the HLA data are more heterogenous and complicated, thus imputation is more difficult. Given the highly correlated and structured genotype data, methods like our SCDA method that take account of these local patterns will work well to impute missing genotypes.

### 2.2. Autoencoders

Autoencoders [62] are unsupervised artificial neural networks that are designed to learn efficient data encoding or representation to reconstruct the original input data. As shown in Figure 2a, an autoencoder consists of two parts: an encoder and a decoder, which can be defined as *f* and *g*, respectively. The encoder takes an input vector x∈ℝn and maps it to a hidden representation h∈ℝm through a mapping function in Equation (1).
(1)h=fθ(x)=Φ(Wx+b),
where θ={W,b}, W is a m×n weight matrix, b is a bias vector, and Φ is an activation function such as a sigmoid [63] or rectified linear units (ReLU) [64]. The hidden representation, h, is also called a latent representation. The decoder takes a hidden representation h to map it to a reconstructed vector *z*
∈ℝn using Equation (2).
(2)z=gθ′(h)=Φ′(W′h+b′),
where θ′={W′,b′}; W′ is a n×m weight matrix; b′ is a bias vector; and Φ′ is an activation function, the same as Φ. The parameters θ and θ′ of an autoencoder will be optimized to minimize the average reconstruction error, as shown in Equation (3).
(3)θ*,θ′*=argmin1n∑i=1nL(x(i),z(i))=argmin1n∑i=1nL(x(i),gθ′(fθ(x(i)))),
where θ*,θ′* are parameters to be learned on data.

The aim of an autoencoder is to reconstruct $z^{\left(i\right)}$ such that $z^{\left(i\right)}\;\approx x^{\left(i\right)}$ by minimizing the loss function $L\left(x^{\left(i\right)},z^{\left(i\right)}\right)$. $L\left(x^{\left(i\right)},z^{\left(i\right)}\right)$ can be defined as the widely used mean squared error for continuous data or cross-entropy for discrete data. In genotype imputation, we minimized the cross-entropy loss between the input x and reconstructed z as defined in Equation (3), as genotype values are discrete.
(4)$$L\left(x^{\left(i\right)},z^{\left(i\right)}\right)=-\left(ylog\left(p\right)+\left(1-y\right)log\left(1-p\right)\right)$$

### 2.3. Denoising Autoencoders

A denoising autoencoder [65] is an extension of a standard autoencoder, which takes corrupted input data with missing values, and can thus be applied for data imputation. A denoising autoencoder reconstructs the output from the corrupted input data by allowing the encoder to extract the most important features and learn a robust representation of the input data, as shown in Figure 2b. When the corrupted input includes missing data values, denoising autoencoders can solve the imputation problem by predicting the values of missing data points by reconstructing the input data [65]. As denoising autoencoders fit well with solving missing data problems, in this project, we utilized denoising autoencoders for genotype imputation. For example, in a synthetic dataset where the yeast data was first corrupted by randomly masking 5% of the original values to zeros, we can train a denoising autoencoder to predict values of those data points masked as zeros.

### 2.4. Sparse Convolutional Networks

However, it is computationally expensive to use solely autoencoders for data imputation, especially when the input data are of large scale, high dimension, and with local structures. Autoencoders are primarily multiple-layer perceptron neural networks with dense layer-wise connections, which can be very expensive to learn in terms of computational time and space. Furthermore, fully connected autoencoders ignore data relationships such as LD and linkage structures in genotype data (Figure 1). To further take advantage of the characteristics of genotype data, we leverage convolution networks to learn the underlying structures and relationships of genotype data. As genotype values are discrete, we use convolution operations in a discrete space defined below.
(5)$$O\left(i\right)=\overset{k}{\underset{u=1}{\sum }}F\left(u\right)I\left(i-u\right),$$
where O(i) is the output of the i marker in the input vector *I*. F is the convolutional filter and k is an odd number representing the convolutional filter size. In this study, we experiment with odd filter sizes ranging from 3 to 19. The convolution operation in Equation (5) is performed for every location of the input vector I, and thus for each genetic marker.

Every convolutional layer is composed of *n* convolutional filters, each with a depth of *D*, where *D* is the input depth. A convolution among an input $I=\left\{I_{1},I_{2},\cdots ,I_{D}\right\}$ and a set of *n* convolutional filters $\left\{F_{1},F_{2},\cdots ,F_{D}\right\}$ produces a set of *n* activation maps or, equivalently, a volume of activation maps with a depth of *n*:(6)$$O_{m}=\sigma \left(I⊗F_{m}+b_{m}\right)\;m=1,\cdots ,n,$$where σ is a non-linear activation function and ⊗ is a convolution symbol of Equation (5). Here, $b_{m}$ is the bias and *m* represents the *m*th feature map.

Overfitting is a critical problem for analyzing highly dimensional data such as genotype data. To prevent overfitting and improve model performance, we introduce the *L*_1_ normalization, defined in Equation (7), to all convolutional filters. The *L*_1_ norm regularization works by applying penalties on layer weights during the optimization process of a model. These penalties are incorporated in the loss function on which the model will optimize. The *L*_1_ norm will penalize or shrink small weights to zeros to improve the robustness of a model.
(7)$$L_{1}=\lambda \overset{n}{\underset{m=1}{\sum }}||F_{m}||{}_{1,1}$$

Here, ||·|| refers to the *L*_1_ norm of a weight matrix, $F_{m}$ is the *m*th convolution filter weight matrix in the layer, and λ∈[0, 1] is a hyperparameter for controlling the model shrinkage or sparsity. The larger λ is, the sparser the trained model will become.

### 2.5. Sparse Convolutional Denoising Autoencoders

To leverage the advantages of denoising autoencoders and convolutional networks, we propose a sparse convolutional denoising autoencoders (SCDA) model for genotype imputation. As the highly correlated LD and linkage patterns are key characteristics in genotype data, we use convolutional networks to incorporate these patterns from input data. Each convolutional kernel generates a feature map from the input, and in this process, LD patterns in the filtering window of the convolutional layer can be incorporated. Moreover, we introduce an *L*_1_ regularization to every convolutional kernel to induce sparsity in the SCDA model.

The proposed network architecture of the SCDA model is shown in Figure 3. There are seven layers in the model, including one input layer and six convolution layers. The input layer takes corrupted data. Each convolution layer is regularized by an *L*_1_ penalty. The number of convolutional kernels in SCDA model is 32, 64, 128, 128, 64, and 1. The hyperparameter of the *L*_1_ norm is set at λ=0.0001.

In the SCDA model, maxpooling and upsampling are used in the model architecture. Maxpooling is down-sample processing to reduce dimensionality, which applies a max filter to non-overlapping subregions of the previous layer. Maxpooling thus reduces the computational cost by reducing the number of parameters and provides basic translation invariance to the internal representation. Upsampling is an opposite processing to maxpooling. Upsampling is used to increase the dimensionality by repeating data along the axis. The filter size of maxpooling and upsampling of SCDA is fixed as 2 [66].

To prevent overfitting, we implemented a commonly-used technique called dropout in the SCDA model. Dropout works by removing neurons and their connected edges either at the hidden or visible layers in a neural network. One simple dropout strategy is that each neuron is kept in the network with a retention probability *p* independent of any other neurons. In this study, we use this dropout strategy to empirically set the dropout probability at 25% [67].

The whole SCDA model was built using TensorFlow v1.13.1 [68] in python3.6, and trained and tested on one NVIDIA GeForce GTX-1080Ti GPU. The batch size is set at 32 and the maximum number of epochs is 1000.

## 3. Results

### 3.1. Optimization of the SCDA Architecture

One of the most critical steps in building a deep learning model is to optimize the model architecture and tune its hyperparameters. These hyperparameters in most deep learning-based approaches are tuned empirically [69], although automatic machine learning methods have been recently proposed [70]. In this study, we empirically tune the hyperparameters of the SCDA model to achieve an optimized SCDA architecture for genotype imputation. Specifically, we first investigated the number of convolutional layers and the convolutional kernel size, as these two hyperparameters usually have large effects on the final performance of a model. One important concern in genotype imputation is to capture the local linkage or correlation patterns that can be incorporated in convolutional layers in a model. The number of convolutional layers in a deep neural network determines the degree of complexity of the relationship among different linkage patterns that a model can learn. The convolutional kernel size determines the local patterns that convolutional layers can capture. We tested two SCDA architectures with five layers and seven layers, and evaluated their performance for genotype imputation respectively. All convolution kernels had the same filter sizes, ranging from 3 to 19. The other hyperparameters were set as canonical values. For an SCDA model with seven layers, the number of kernels of the six convolution layers was 32, 64, 128, 128, 64, and 1. For an SCDA model with five layers, the number of kernels of the four convolution layers was 32, 64, 32, and 1. Dropout filter size was fixed at 25%, and maxpooling and upsampling filter sizes were set at 2.

These optimizations were performed on the yeast genotype dataset that was randomly corrupted at a 10% missing level. As shown in Figure 4, the seven-layered architecture with the filter size of five achieved the best performance with an accuracy of 0.9986, followed by the five-layered architecture with the filter size of five, with an accuracy of 0.9983. We observed that the filter size of five is the best convolutional filter size in both five-layered and seven-layered SCDA architectures.

Another important hyperparameter in the SCDA model is the number of convolutional kernels. The more kernels in a layer, the more patterns a convolutional network can capture. However, the number of kernels chosen also depends on the complexity of data. The number of convolutional kernels at a later layer is expected to be larger than that of the previous layer, as the number of possible combinations grows. Here, we increased the number of convolutional kernels by doubling it in the encoder, and reducing it to half in the decoder. We evaluated three combinations (16, 32, 64, 64, 32, 1), (32, 64, 128, 128, 64, 1), and (48, 96, 192, 192, 96, 1). The results in Figure 5 show that the combination with (32, 64, 128, 128, 64, 1) achieved the best performance.

Hence, we finalized our SCDA model for genotype imputation as a network infrastructure with seven layers with the number of kernels (32, 64, 128, 128, 64, 1) and a fixed filter size of five. The kernel visualization of first convolution layer of the SCDA model with seven layers is shown in Figure 6. We observed that more than half of the weight values are zeros (denoted as grey squares), indicating the sparsity of our model.

### 3.2. The Performance and Robustness of the SCDA Model

To evaluate the performance and robustness of our SCDA model, applied the model to the yeast data and the HLA data to impute genotypes in three different missing levels at 5%, 10%, and 20%, respectively. We trained the SCDA models for 10 times by randomly splitting the data into training, validation, and testing datasets with 65%, 15%, and 20% of the data at each missing level. The imputation performance of our SCDA model is shown in Table 2. For yeast genotype imputation, SCDA has an average accuracy of 0.9978, 0.9977, and 0.9975, respectively, with a standard deviation of 7.0 × 10^−5^, 3.9 × 10^−5^, and 7.0 × 10^−5^, respectively, at missing scenarios with 5%, 10%, and 20% missing values, respectively. Hence, the SCDA achieved comparable performance for the three missing scenarios, which indicates that SCDA works for imputing noisy data where a large proportion of the data is missing (e.g., 20% of missing values). Nonetheless, as expected, its performance is higher when the missing level is lower and the missing data is easier to impute. The low standard deviations indicate that the SCDA model is robust in these scenarios.

As previously reported, populations have a strong effect on human genotype imputation [71]. Therefore, we tested our proposed SCDA model on HLA genotypes of the EAS super population and the entire five super populations worldwide, respectively. For the HLA genotype imputation of the EAS super population, SCDA has an average accuracy of 0.9975, 0.9952, and 0.9942, respectively, with a standard deviation of 6.0 × 10^−5^, 1.4 × 10^−4^, and 4.2 × 10^−4^, respectively, at three missing scenarios with 5%, 10% and 20% missing values, respectively. For the HLA genotype imputation on the entire super populations, SCDA has an average accuracy of 0.9973, 0.9949, and 0.9896, respectively, with a standard deviation of 1.9 × 10^−^4, 7.5 × 10^−^5 and 1.5 × 10^−4^, respectively, at three missing scenarios with 5%, 10%, and 20% missing values, respectively. As expected, the performance of genotype imputation on a single EAS population is better than that on the entire human population including five super populations. This is because of the fact that human genotypes worldwide are more heterogeneous and complicated across diverse populations, and it is more difficult to capture local LD patterns in genotype imputation.

Compared with the yeast data, the correlation patterns in HLA genotypes are more dispersed and heterogeneous (Figure 1), and thus the imputation of human HLA genotypes is more difficult than that of yeast genotypes. SCDA achieved an overall average accuracy of 0.9977 for yeast genotypes, which is better than the overall average accuracy of 0.9942 for HLA genotypes of EAS and an overall average accuracy of 0.9939 for entire HLA genotypes.

### 3.3. Comparison with Other Methods

We compared our reference-free genotype imputation methods, SCDA, to other reference-free imputation methods on these datasets. Particularly, we chose three commonly used statistical inference algorithms for data imputation including row average [24], KNN [25], and SVD [26] for comparison. The average accuracy and standard deviation of each imputation method were calculated by performing the method 10 times at every missing scenario (5%, 10%, and 20%). The total average and standard deviation were calculated by averaging the results of three missing scenarios.

The imputation results on the yeast data summarized in Table 3 show that the SCDA model achieves a total average accuracy of 0.9977, significantly outperforming the other imputation methods in comparison. KNN and SVD have comparable performance, which performed much better than the simplest strategy of row average imputation. SCDA also has the lowest standard deviation, which means that it is the most robust method compared with the other three imputation methods. The results on imputing HLA genotypes, described in Table 4 and Table 5, show that the SCDA model outperforms the other three imputation methods in comparison as well, achieving the total average accuracy of 0.9942 and 0.9939 for the EAS super population and the entire human population, respectively.

We noticed that the methods in comparison, namely the average, KNN, and SVD methods, all achieved much better imputation performance on HLA genotypes than yeast genotypes, as they did not take into account the LD or correlation structures most obvious in the yeast data. Our SCDA model achieved comparable performance on the yeast and HLA genotypes, which indicates SCDA has strong robustness in imputing genotypes from homologous or heterogeneous population backgrounds.

These imputation results on the yeast and HLA genotypes were also visualized in violin plots in Figure 7 and Figure 8. Higher median point indicates higher performance, and tighter distribution indicates greater robustness. We observed that SCDA has the highest median points and tightest distributions on both yeast and HLA data, which demonstrates that our SCDA model achieved the state-of-the-art performance compared with other methods for data imputation.

## 4. Discussion

In summary, we presented a novel deep learning model called SCDA for genotype imputation based on sparse convolutional denoising autoencoders. This SCDA model achieves state-of-the-art imputation accuracy compared with popular reference-free imputation methods. Additionally, the SCDA model is robust in different levels of missing data and heterogeneity of genotype data, making it a competing method for genotype imputation. The nice performance of our SCDA model benefits from its multiple convolutional layers that can extract hidden data patterns and its sparse architecture due to the added *L*_1_ regularization on the weight matrix.

In future, we will apply the SCDA model to more complex datasets with untyped genotypes in real scenarios. As the SCDA is based on a deep learning architecture, it is a computationally demanding process to train the model and many hyperparameters are set empirically. We will adopt more efficient training mechanisms and explore more comprehensive and automatic hyperparameter learning. The SCDA model also suffers from the common weakness of deep learning models in that it is hard to explain the prediction mechanisms. We will add prior domain knowledge and provide network visualization in the future to mitigate this limitation of SCDA towards explainable artificial intelligence.

All in all, this study demonstrates another new application of deep learning to the problem of missing data imputation in genomic studies. Although we originally designed the SCDA model for genotype imputation, our SCDA model can be applied to infer missing values in any data matrix including high-dimensional matrices and tensors. The genotype values we use here are discrete values, but the SCDA model can be extended to impute other kinds of missing values, including quantitative values in gene expression or DNA methylation data. The current SCDA architecture can be extended to multi-task imputation to infer missing values in multiple data sets. This deep learning architecture will thus be of great use in data integration such as omics data integration and imaging genomics.

## Figures and Tables

**Figure 1 genes-10-00652-f001:**
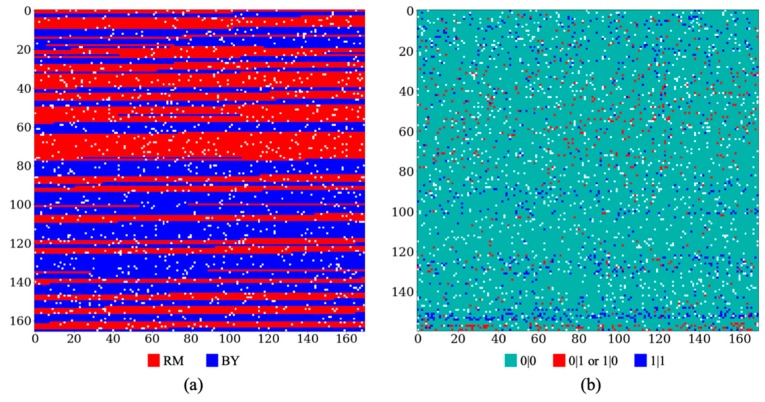
The visualization of genotype profiles of two samples with 5% missing values in (**a**) yeast (BY represents genotypes from a laboratory strain and RM stands for genotypes from a vineyard strain) and (**b**) human leukocyte antigen (HLA) datasets shows the different correlated patterns in the data. Synthetically generated missing values are denoted in white color, while typed genotypes are color coded.

**Figure 2 genes-10-00652-f002:**
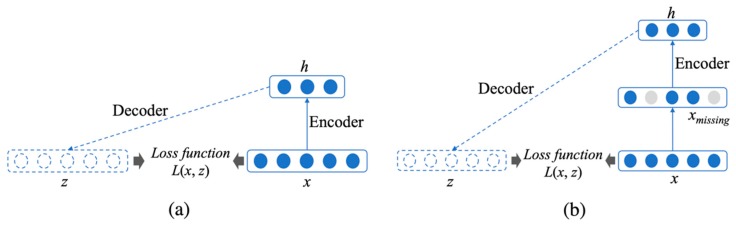
An illustration of a (**a**) standard autoencoder and (**b**) denoising autoencoder. An autoencoder is composed of two parts: encoder and decoder. The encoder takes an input vector *x* and maps it to a hidden representation *h*. The decoder takes a hidden representation *h* to map it to a reconstructed vector *z*. The aim of an autoencoder is to generate a reconstruction *z* of the input data such that z ≈x by minimizing the loss function L(x,z). A denoising autoencoder differs from a standard autoencoder in that the input *x* is corrupted with noises or missing values.

**Figure 3 genes-10-00652-f003:**
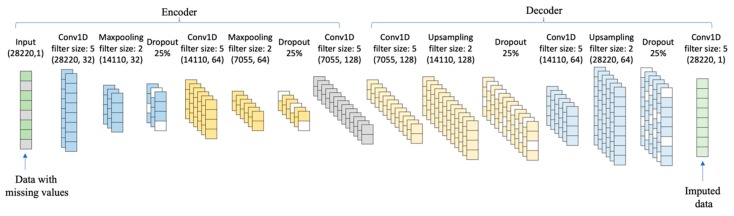
The network architecture of SCDA. The sparse convolutional denoising autoencoders (SCDA) model consists of multiple convolution layers in a hierarchical way, and each convolution layer is regularized by an *L*_1_ penalty. SCDA takes corrupted data with missing values as input to learn a hidden representation, and then reconstructs the input based on hidden representations to impute missing values.

**Figure 4 genes-10-00652-f004:**
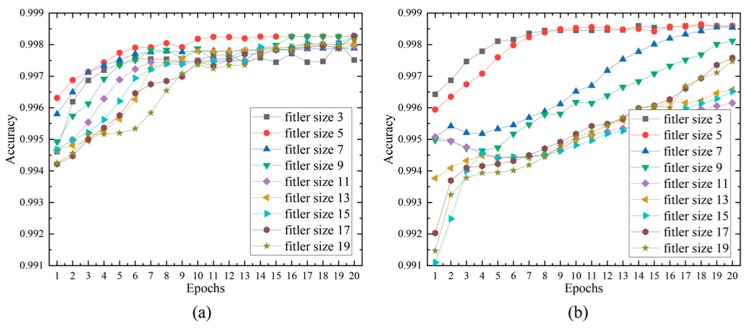
Results of using the SCDA model on the yeast data. (**a**) The results of the five-layered SCDA with filter sizes ranging from 3 to 19. (**b**) The results of the five-layered SCDA with filter sizes ranging from 3 to 19.

**Figure 5 genes-10-00652-f005:**
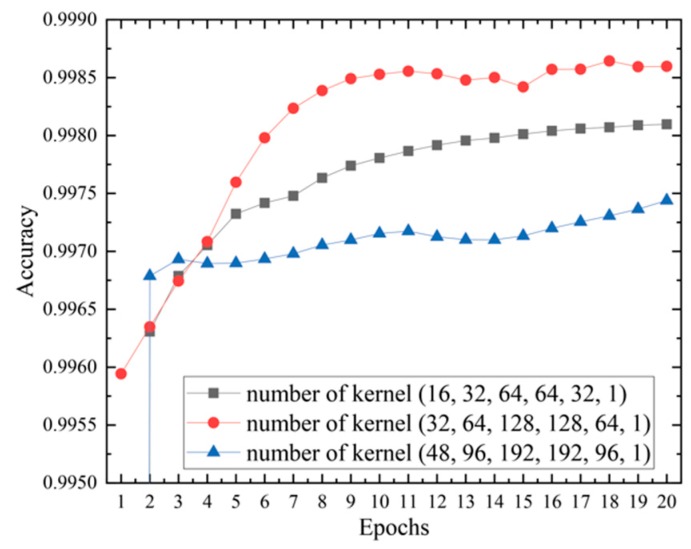
Results of SCDA with different number of kernels on yeast data. The kernel combination with (32, 64, 128, 128, 64, 1) achieves the best performance.

**Figure 6 genes-10-00652-f006:**
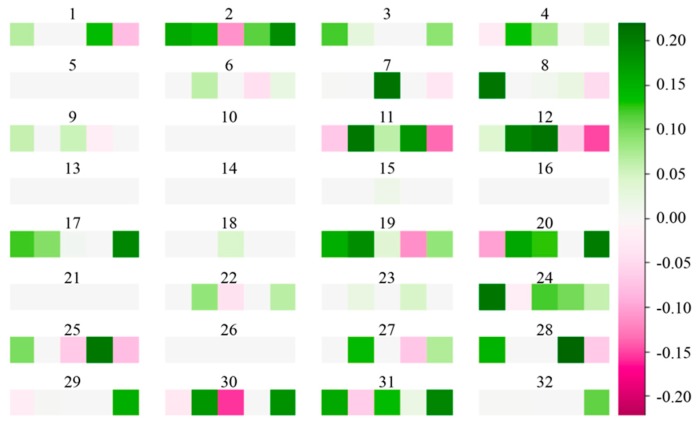
Kernel visualization in the first convolution layer of the SCDA model. There are 32 kernels in the first convolution layer, and each kernel has five squares. The squares in green indicate positive weights, and those in magenta represent negative weights. The deeper the color, the larger the absolute value of the weight. The grey squares represent weights with zero values.

**Figure 7 genes-10-00652-f007:**
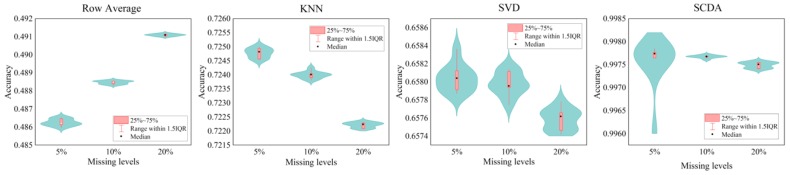
Violin plots of the accuracy values of different imputation methods on yeast genotypes at three missing levels. KNN, k-nearest neighbors; SVD, singular value decomposition.

**Figure 8 genes-10-00652-f008:**
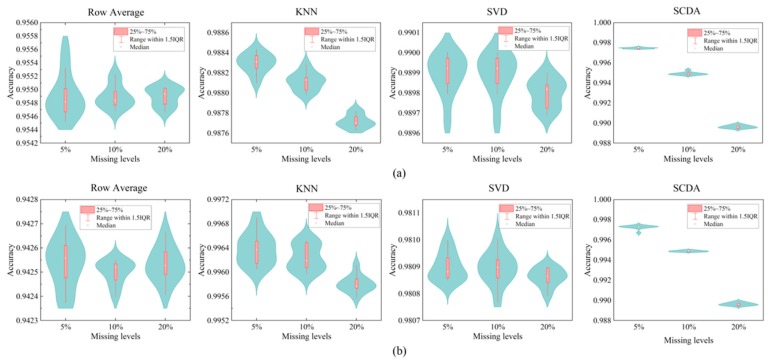
Violin plots of the accuracy values different imputation methods on HLA genotypes at three missing levels. (**a**) Imputation results on HLA genotypes of EAS super population. (**b**) Imputation results on HLA genotypes of the entire human population including five super populations.

**Table 1 genes-10-00652-t001:** Summary of existing genotype imputation methods. HMM, hidden Markov model; SNP, single nucleotide polymorphism; KNN, k-nearest neighbors; SVD, singular value decomposition.

Categories	Methods	Algorithms	References
Reference panel required	fastPHASE	Haplotype-cluster	[7]
	IMPUTE	HMM	[8]
	IMPUTE2	HMM	[11]
	MACH	HMM	[12]
	BEAGLE	Graphical model	[17,18]
	PLINK	Tag SNP	[19]
	SNPMSTAT	Tag SNP	[20]
Reference panel-free	Row average	Mean value	[24]
	Nearest Neighbors	KNN	[25]
	Matrix decomposition	SVD	[26]
	Regression prediction	Logistic regression	[27]
	Classification prediction	Random Forest	[28]

**Table 2 genes-10-00652-t002:** Performance comparison in terms of accuracy on yeast and human leukocyte antigen (HLA) genotypes at three missing scenarios. The average accuracy and standard deviation of each imputation method was calculated by running the sparse convolutional denoising autoencoders (SCDA) model 10 times at every missing scenario (5%, 10%, 20%). The overall average and standard deviation values were calculated by averaging the results of three missing scenarios.

	5%		10%		20%		Total	
	Average Accuracy	Standard Deviation	Average Accuracy	Standard Deviation	Average Accuracy	Standard Deviation	Average Accuracy	Standard Deviation
Yeast	0.9978	$7.0\times 10^{-5}$	0.9977	$3.9\times 10^{-5}$	0.9975	$7.0\times 10^{-5}$	0.9977	$6.0\times 10^{-5}$
HLA_EAS ^1^	0.9975	$6.0\times 10^{-5}$	0.9952	$1.4\times 10^{-4}$	0.9900	$4.2\times 10^{-4}$	0.9942	$2.1\times 10^{-4}$
HLA_Entire ^2^	0.9973	$1.9\times 10^{-4}$	0.9949	$7.5\times 10^{-5}$	0.9896	$1.5\times 10^{-4}$	0.9939	$1.4\times 10^{-4}$

^1^ HLA genotypes of EAS super population from 1000 Genome Project. ^2^ HLA genotypes from the entire dataset in five super populations in the 1000 Genome Project.

**Table 3 genes-10-00652-t003:** Performance comparison on yeast genotypes at three missing scenarios.

Methods	5%		10%		20%		Total	
	Average Accuracy	Standard Deviation	Average Accuracy	Standard Deviation	Average Accuracy	Standard Deviation	Average Accuracy	Standard Deviation
Average	0.4862	1.9 × 10^−4^	0.4885	9.5 × 10^−5^	0.4911	6.7 × 10^−5^	0.4886	1.2 × 10^−4^
KNN	0.7248	1.7 × 10^−4^	0.7240	1.2 × 10^−4^	0.7222	9.4 × 10^−5^	0.7237	1.3 × 10^−4^
SVD	0.6580	1.2 × 10^−4^	0.6580	1.1 × 10^−4^	0.6576	1.1 × 10^−4^	0.6579	1.3 × 10^−4^
SCDA	0.9978	7.0 × 10^−5^	0.9977	3.9 × 10^−5^	0.9975	7.0 × 10^−5^	0.9977	6.0 × 10^−5^

**Table 4 genes-10-00652-t004:** Performance comparison on HLA genotypes of EAS super population at three missing scenarios.

Methods	5%		10%		20%		Total	
	Average Accuracy	Standard Deviation	Average Accuracy	Standard Deviation	Average Accuracy	Standard Deviation	Average Accuracy	Standard Deviation
Average	0.9549	3.2 × 10^−4^	0.9549	1.7 × 10^−4^	0.9549	1.3 × 10^−4^	0.9549	2.1 × 10^−4^
KNN	0.9883	9.5 × 10^−5^	0.9881	8.8 × 10^−5^	0.9877	6.4 × 10^−5^	0.9880	8.2 × 10^−5^
SVD	0.9899	1.0 × 10^−4^	0.9899	9.0 × 10^−5^	0.9898	6.9 × 10^−5^	0.9899	8.6 × 10^−5^
SCDA	0.9975	6.0 × 10^−5^	0.9952	1.4 × 10^−4^	0.9900	4.2 × 10^−4^	0.9942	2.1 × 10^−4^

**Table 5 genes-10-00652-t005:** Performance comparison on HLA genotypes of the five super populations at three missing scenarios.

Methods	5%		10%		20%		Total	
	Average Accuracy	Standard Deviation	Average Accuracy	Standard Deviation	Average Accuracy	Standard Deviation	Average Accuracy	Standard Deviation
Average	0.9498	9.6 × 10^−5^	0.9497	5.8 × 10^−5^	0.9498	4.5 × 10^−5^	0.9498	6.6 × 10^−5^
KNN	0.9873	5.8 × 10^−5^	0.9871	4.3 × 10^−4^	0.9867	3.5 × 10^−5^	0.9870	4.5 × 10^−5^
SVD	0.9809	9.2 × 10^−5^	0.9809	4.4 × 10^−5^	0.9809	2.9 × 10^−5^	0.9809	5.5 × 10^−5^
SCDA	0.9973	1.9 × 10^−4^	0.9949	7.5 × 10^−5^	0.9896	1.5 × 10^−4^	0.9939	1.4 × 10^−4^

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
