# Peer review of "Sparse Convolutional Denoising Autoencoders for Genotype Imputation"

_genes, 2019, doi:10.3390/genes10090652_

Round 1

Reviewer 1 Report

The authors are to be congratulated for developing a very useful genotype imputation approach that does not need a high-resolution reference.  Missing data is serious problem in genetics, and this approach is not only useful for imputing missing genotypes but also can be extended to quantitative gene expression and methylation data.  I have one minor question.  How might the Lregularization compare to say the combination of the L1 and Lunder the "Elastic Net Regularization" approach?

Author Response

Reviewer #1

The authors are to be congratulated for developing a very useful genotype imputation approach that does not need a high-resolution reference.  Missing data is serious problem in genetics, and this approach is not only useful for imputing missing genotypes but also can be extended to quantitative gene expression and methylation data.  I have one minor question.  How might the Lregularization compare to say the combination of the L1 and Lunder the "Elastic Net Regularization" approach?

A: We tried the Elastic Net Regularization   in which L1 and L2 have the same weights, and , same as in the L1 regularization used in our SCDA model. The Elastic Net Regularization approach achieved similar performance with L1 regularization. We thus chose to use L1 regularization used in our SCDA model for simplicity.

Reviewer 2 Report

Junjie et al. proposed a deep-learning model called SCDA for imputing the missing genotype. This model used convolutional layers to consider the correlations in genotype data, and applied L1 regularization to deal with the sparsity of the model on each convolutional kernel. By using the simulated missing data in yeast genotype dataset and human HLA genotype dataset, SCDA demonstrated its better performance than other methods (row average, KNN, and SVD). This manuscript is generally well written.

Specific Comments:

1.     As this paper is proposing a computational model, it is very necessary to provide the codes as an open source software (e.g. Github or Gitlab), or a webserver, or at least a supplementary document. The codes should be well commented. The authors should do so, to allow other users to apply this model, and probably improve this model as well. Without the codes, this model is impractical.

2.     The authors did not mention which deep-learning framework was used to build their model. The technical details are needed, including the framework used, the hardware configurations in running the model, and the time cost.

3.     As the authors only used simulated missing data in the model performance evaluation, is there any real examples can be used to test the model in genotype imputation?

4.     Line 297: What is the ‘empirical fashion’. It needs a citation and an explanation. This question is also related with the following questions regarding the parameters.

5.     Line 306: Why only testing the SCDA architectures with 5 and 7 layers only?

6.     Line 309: Why using (32, 64, 128, 128, 64, 1) kernels in the 7-layer model, and (32, 64, 32, 1) kernels in the 5-layer model?

7.     Line 310: Why fixing dropout filter size as 25%, and maxpooling and upsampling filter size as 2?

8.     Figure 7 & figure 8: organize these two figures in the same layout, meaning line the four different methods on the same row, for figure 8.

9.     Any weakness of SCDA model, compared with other methods?

Author Response

Junjie et al. proposed a deep-learning model called SCDA for imputing the missing genotype. This model used convolutional layers to consider the correlations in genotype data, and applied L1 regularization to deal with the sparsity of the model on each convolutional kernel. By using the simulated missing data in yeast genotype dataset and human HLA genotype dataset, SCDA demonstrated its better performance than other methods (row average, KNN, and SVD). This manuscript is generally well written.

Specific Comments:

As this paper is proposing a computational model, it is very necessary to provide the codes as an open source software (e.g. Github or Gitlab), or a webserver, or at least a supplementary document. The codes should be well commented. The authors should do so, to allow other users to apply this model, and probably improve this model as well. Without the codes, this model is impractical.

A: We have released our code in GitHub via https://github.com/shilab/SCDA

The authors did not mention which deep-learning framework was used to build their model. The technical details are needed, including the framework used, the hardware configurations in running the model, and the time cost.

A: Technical details are added in section 2.4.

As the authors only used simulated missing data in the model performance evaluation, is there any real examples can be used to test the model in genotype imputation?

A: We didn’t use real missing samples to test the model, since the yeast data we use is comprehensively genotyped with no missing values. The human HLA genotype data has missing values, however, we don’t know the real genotypes of those missing values and thus cannot evaluate the performance of our imputation methods. To test the robustness of our model, we hence simulated three missing scenarios by randomly masking 5%, 10% and 20% of the original genotypes, which cover common situations in genotype imputation. As suggested, we will improve our model and test it on real examples using different datasets in the future.

Line 297: What is the ‘empirical fashion’. It needs a citation and an explanation. This question is also related with the following questions regarding the parameters.

A: Empirical fashion means that we tuned the hyperparameters manually based on previous empirical evidence. Since the training process in deep learning is usually computing and time intensive, empirical strategy is commonly used to optimize hyperparameters [1].

Bergstra, J.; Bengio, Y. Random search for hyper-parameter optimization. Journal of Machine Learning Research 2012, 13, 281-305.

Line 306: Why only testing the SCDA architectures with 5 and 7 layers only?

A: The hyperparameters were chosen empirically to balance the performance and complexity of the model. We tested several architectures. As Figure 4 shown, the 7-layered architecture is slightly better than the 5-layered architecture. We only tested the SCDA architectures with 5 and 7 layers since more layers can’t significantly improve the performance while the computing cost is increased dramatically.

Line 309: Why using (32, 64, 128, 128, 64, 1) kernels in the 7-layer model, and (32, 64, 32, 1) kernels in the 5-layer model?

A: SCDA is a full CNN-based autoencoder model. In general, the larger the number of kernels in a layer, the more patterns it can capture. However, the number of kernels used in a model also depends on the complexity of data. As the number of possible combinations grows, the number of kernels at a later layer is expected to be larger than that of the previous layer. That is why, in general, the first layer has fewer kernels than hidden layers. We increased the number of convolutional kernels by doubling it for an encoder, and reduced it to half for a decoder. We tested several kernel combinations with more kernels in the 7-layer model. The results in Figure 5 show the kernel combination that the first layer with 32 kernels achieves the best performance.

Line 310: Why fixing dropout filter size as 25%, and maxpooling and upsampling filter size as 2?

A: These hyperparameters were set empirically based on previous studies [1,2].

Radford, A., L. Metz and S. Chintala (2015). "Unsupervised representation learning with deep convolutional generative adversarial networks." arXiv preprint arXiv:1511.06434. Achille, A. and S. Soatto (2018). "Information dropout: Learning optimal representations through noisy computation." IEEE transactions on pattern analysis and machine intelligence 40(12): 2897-2905.

Figure 7 & figure 8: organize these two figures in the same layout, meaning line the four different methods on the same row, for figure 8.

A: This issue has been fixed.

Any weakness of SCDA model, compared with other methods?

A: Since SCDA is a deep learning-based model, it needs a long time to train to achieve a good performance. Additionally, the common weakness of deep learning-based model is that it is hard to explain the prediction mechanisms. We have added these points to the discussions of the manuscript.

Round 2

Reviewer 2 Report

Thanks for addressing the comments, revising the manuscript and providing a GitHub project link. However, the GitHub project only provides the installation steps, lacks the description and the case study to demonstrate its use for practical purposes. Prediction models are built for helping others' research in relevant fields, not for showing the performance in a paper only.

Author Response

Thanks for addressing the comments, revising the manuscript and providing a GitHub project link. However, the GitHub project only provides the installation steps, lacks the description and the case study to demonstrate its use for practical purposes. Prediction models are built for helping others' research in relevant fields, not for showing the performance in a paper only.

A: We have updated the GitHub project. A case study for yeast genotype imputation was added ( in README.md) to demonstrate the usage of our SCDA model for practical purposes. We provided the training (SCDA_train.ipynb) and testing (SCDA_test.ipynb) code in Jupyter Notebooks, which can be easily run by users in a step-by-step manner. These Jupyter Notebooks can be easily converted to python scripts. One trained model on the yeast dataset is provided. In future, we will train more models in various species where users can use these trained models to directly impute missing genotypes on the genomes of interest.